# Facing the Increased Prevalence of Antibiotic-Resistant *M. tuberculosis*: Exploring the Feasibility of Realising Koch’s Aspiration of Immunotherapy of Tuberculosis

**DOI:** 10.3390/antibiotics11030371

**Published:** 2022-03-10

**Authors:** Peter A. Bretscher

**Affiliations:** Department of Biochemistry, Microbiology and Immunology, University of Saskatchewan, Heath Sciences Building, 105 Wiggins Road, Saskatoon, SK S5N 5E5, Canada; peter.bretscher@usask.ca

**Keywords:** tuberculosis, host-targeted therapies, immunotherapy

## Abstract

Koch attempted to treat tuberculosis in the late 1800s by administering an antigenic extract derived from the pathogen to patients. He hoped to bolster the patient’s protective immunity. The treatment had diverse results. In some, it improved the patient’s condition and in others led to a worsening state and even to death. Koch stopped giving his experimental treatment. I consider here three issues pertinent to realizing Koch’s vision. Rational immunotherapy requires a knowledge of what constitutes protective immunity; secondly, how on-going immune responses are regulated, so the patient’s immunity can be modulated to become optimally protective; thirdly, a simple methodology by which treatment might be realized. I deliberately cast my account in simple terms to transcend barriers due to specialization. The proposed immunotherapeutic treatment, if realizable, would significantly contribute to overcoming problems of treatment posed by antibiotic resistance of the pathogen.

## 1. The Immunology of Tuberculosis Is Complex

Rational approaches to successful vaccination and immunotherapy of any infectious disease are greatly facilitated by a delineation of the correlates that discriminate the immunity of the healthy infected, who contain the pathogen, and of those who suffer disease. Such a clear delineation, if possible, allows investigators to infer what kind of immunity is protective. Discriminatory immunological correlates are particularly difficult to find in those infected with *Mycobacterium tuberculosis* [1,2,3]. Although many observations suggest that a predominant Th1 response is protective, there are, for a start, two observations we should consider when attempting to develop an integrated picture concerning the role of immunity in both protection and pathogenesis. Lung granulomas are pathological in the most common form of lung tuberculosis. Pathogen specific Th1 cells contribute to granuloma formation [1]. The first question is: what different circumstances can explain how Th1 cells are protective but also contribute to pathology? Secondly, it is also known that the humoral/cell-mediated nature of the immune response in patients with active tuberculosis has a very wide spectrum; at one extreme are patients with miliary tuberculosis, who produce substantial amounts of IgG antibodies, and often express poor delayed-type hypersensitivity (DTH) [4]; at the other extreme are patients who express DTH but produce little antibody. Patients, *on average*, have *higher levels* of IgG antibody than the healthy infected; this, however, is not a “clean observation”, as discussed below; many *patients* have *lower* levels of IgG antibody than many of the healthy infected [2]. These observations cannot be reconciled with there being simple immunological correlates discriminating between responses that are protective from those leading to disease [3].

It is essential for the rational design of immunotherapy to know the nature of the immunity that is protective. In face of the fact that there are no clear correlates distinguishing the immunity of patients from immunity in the healthy infected, what information might be helpful? Evidence showing that modulation of the immune response, in a certain direction, results in consistent and reliable resistance to a challenge that causes disease in genetically diverse and naïve animals would be most helpful. Buddle has found a way of vaccinating cattle so they dramatically and reliably resist an otherwise lethal challenge of *Mycobacterium bovis* [5]. This significant study is only rarely cited in the literature, even in those situations where I think it pertinent. It may provide in part a foundation we are all seeking in understanding the role of immunity in tuberculosis and for designing immunotherapy of this disease. It is my purpose in this article to explore this avenue.

## 2. An Appeal to the Reader

I have been an immunologist for over 50 years. I started out as a basic immunologist, particularly interested in trying to understand the basis of self-nonself discrimination [6] and immune class regulation [7]. I always took a broad interest in those fields of immunology that are related to medicine, particularly autoimmunity, infectious disease, and cancer. This was not so difficult 50 years ago, as the fields were much smaller and more intimate than they are today. My involvement in the field for so many years means I am aware of older studies from the 1960s–1990s. These older studies are, in my experience, rarely known by my younger colleagues, and there are few colleagues left of my vintage. I believe that the intensity of research has changed, due to the information overload, so that researchers must often specialize to survive. There is little research interaction between people in neighboring fields, and so research silos are prevalent. I appeal to readers to consider whether my thoughts, often triggered by some findings outside and by others within the field of immunity against tuberculosis, make sense. I do not ignore findings in the more focused field, but I have realized I cannot address all the queries that might be made by readers who are specialists. I found a relatively recent review on the problems in understanding the nature of protective immunity against *M. tuberculosis* [1] very incisive and inspiring. I and my colleagues wrote a paper on our ideas on the nature of protective immunity, based upon observations we had made [3]. This review provided our inspiration for addressing the paradoxes of the field so concisely laid out there. I cannot recapitulate all our considerations here that we expressed in our paper on the nature of protective immunity. Rather, I would like to briefly indicate how a broad perspective may have something to offer.

## 3. The Context for Efficacious Vaccination of Cattle against Tuberculosis

Buddle, in describing his study [5], cited ours on vaccinating mice against the human pathogen responsible for cutaneous leishmaniasis [8]. Cutaneous leishmaniasis appears to be much simpler, immunologically speaking, than tuberculosis, in that the correlates discriminating between the immunity of the healthy infected and of patients seems to be unequivocally clear [9]. In brief, predominant Th1 responses appear to be protective and the anti-pathogen immunity in patients has a significant or predominant Th2 component. In this case, it seems clear what vaccination must achieve. Vaccination must cause a Th1-imprint that ensures a strong and predominant Th1 response upon natural infection. We achieved this in the mouse model of cutaneous leishmaniasis [8,10]. Our approach was based on the studies of others which I briefly describe to provide context.

## 4. The Basis for the Low Dose Vaccination Strategy against Cutaneous Leishmaniasis

A classic study by Salvin of the late-1950s delineated how the dose of antigen, and time after immunization, affects the cell-mediated/IgG nature of the response [11]. Salvin measured DTH to assess cell-mediated immunity, and IgG antibodies to assess the humoral component of the response. His observations are summarised in Figure 1. A sufficiently low dose of antigen induces only cell-mediated immunity. A medium dose induces an exclusive cell-mediated response more quickly, but this response declines as IgG antibodies are produced. Even higher doses of antigen induce more rapid responses, and the cell-mediated phase may be even eclipsed. These observations have been confirmed in diverse studies.

Parish showed in the late 1960s that repetitive immunization over weeks with a low dose of antigen not only generates a sustained DTH response but causes a DTH imprint. Rats so exposed were given a larger challenge of antigen that in unexposed rats generated a robust IgG antibody response with little expression of DTH. The pre-exposed rats expressed DTH and produced less IgG antibody than their naïve counterparts to the challenge [12].

Starting in the late 1980s, Coffman and Mossman began publishing their studies establishing that there were two major subsets of Th cells, Th1 cells that produce IFN-gamma and mediate DTH, and Th2 cells that produce IL-4 and facilitate the production by B cells of IgG (or more precisely IgG_1_) antibody [13]. These statements are deliberately simplified to express only the most essential, not the complete, findings.

Howard, in the early 1980s, established a mouse model of human cutaneous leishmaniasis [14]. He and colleagues infected different strains of mice with 10^6^
*Leishmania major* parasites. The susceptible BALB/c strain of mouse produces an IgG antibody and suffered parasitemia and progressive disease, whereas most other strains generate a sustained DTH response and are resistant, associated with control of the parasites. It became apparent in the late 1980s that resistance and susceptibility were correlated with the generation of Th1 and Th2 cells, respectively [15]. This correlation was recognized over time as likely being more than correlative, and as most probably being causal. It was found that a manipulation that modulated the response of susceptible mice from a Th2 to a Th1 phenotype rendered the mice resistant [16,17] and, conversely, a manipulation that modulated the response of resistant mice from a Th1 to Th2 phenotype made them susceptible [18].

Given all these developments, we sought to explore whether Parish’s approach to achieving DTH imprinting to a protein antigen, as outlined above, could be exploited as a means of establishing Th1 imprints in susceptible, BALB/c mice, rendering them resistant against the parasite. We reported in 1992 that infection with 300 parasites resulted in a stable Th1 response, and a Th1 imprint. These mice, challenged with 10^6^ parasites two months after their exposure to 300 parasites, generated a stable Th1 response and were resistant [8,10]. We naturally refer to our strategy as the low dose vaccination strategy.

## 5. Coherence of Immune Responses

It is natural to recognize that there are target antigens of a pathogen against which immunity is required to contain the pathogen. A natural approach to achieving effective immunity is to start by defining these antigens. Is such a definition necessary?

It is remarkable that the CD4 T cells which are specific for diverse antigens of a pathogen, under particular circumstances of infection, often belong to the same subset. Thus, infection with low numbers of mycobacteria or leishmania parasites results in all the activated CD4 T cells belonging to the Th1 but not the Th2 subset. This coherence in the response is partially explained by the recognized ability of Th cells belonging to one subset of inhibiting the generation of Th cells belonging to another Th subset. I work on the assumption that coherence is a general feature of immune responses. Thus, if we can find a way of getting the correct type of protective immunity against all the antigens of the pathogen, we will have the effective immunity to the target antigens. Our and Buddle’s studies on vaccination against mycobacteria and leishmania parasites show that vaccination strategies can be developed without defining “target antigens.” This is not to imply that vaccination with target antigens should not be explored or employed if found to be effective.

## 6. Buddle’s Study

The Bacillus Calmette–Guérin (BCG) vaccine is an attenuated strain of *M. bovis* that has been used for vaccinating cattle and people against tuberculosis. The efficacy of this vaccine is variable and complex, being partially protective against some form of tuberculosis in some people living in certain geographical areas [19]. However, it does not generally and reliably result in protection. Given the prevalence of tuberculosis, BCG has been employed more than any other vaccine in attempts to control the spread of the disease [19].

Buddle reported in 1995 that he could very robustly protect some cattle against an experimental and pathogenic challenge of *M. bovis*. Remarkably, he achieved such protection by reducing the dose of BCG employed in previous studies by about a million-fold [5].

## 7. Can the Low Dose Vaccination Strategy Be Effective in a Genetically Diverse Population?

Our success in vaccinating susceptible BALB/c mice against *L. major* led us to consider a major problem in developing a universally efficacious vaccine against an invader preferentially susceptible to cell-mediated attack. We know that genetic diversity of the human or any animal population means that the same vaccination protocol often results in different kinds and intensities of immune responses in different individuals. This genetic diversity would also in principle be a problem in generating universally efficacious vaccination that guarantees a larger/more rapid antibody response on natural infection. However, priming with a sufficiently high dose of antigen will generate in all individuals a memory state that results, upon infection, in such a response in all vaccinated individuals. Therefore, current vaccines can be effective in the very large majority of individuals. However, it was not obvious to us how the low dose vaccination strategy could be applied to result in universally efficacious vaccination in a genetically diverse population.

If a universally efficacious protocol is to be developed, it must depend on a generalization that holds in all individuals. We thought it would be worthwhile to explore whether the dependence of the Th1/Th2 phenotype of the response, on antigen/parasite dose, was broadly valid. We therefore infected mice of diverse strains with different numbers of parasites and monitored the Th1/Th2 phenotype of the ensuing responses. We employed the same parasite strain and protocol of infection in all our studies, just varying the number of parasites injected. We found that infection with relatively low numbers of parasites resulted in a stable Th1 response, whereas infection with higher numbers resulted in responses that in time had a substantial or predominant Th2 component. We could define for any mouse strain a “transition number”, N_t_; infection with a number below N_t_ results in a stable Th1 response, and with a number above N_t_ results in time in a response with a substantial Th2 component. The greater the number of parasites injected above N_t_ the more rapidly did a substantial Th2 component of the response develop [20], as might be anticipated from the observations depicted in Figure 1. This ability to define N_t_, in all strains of mice tested, supports the generality of the dose dependency of the Th1/Th2 phenotype of the ensuing response.

The value of the transition number varied among different strains of mice over a 100,000 fold range [20]! This initially surprised us. On reflection, we appreciated that genetic diversity is likely very important for our survival as a species. Consider HIV-1 infection. About 1% of naturally infected people generate a stable Th1 response and are the elite controllers [21,22]. All the other infected individuals, if untreated, would seroconvert, develop AIDS and die. Thus, our genetic diversity seems important in protecting us as a species against some newly arising pathogens.

We can rationalize how immunization can be universally efficacious against a pathogen contained by IgG antibody. Priming with a number of attenuated parasites/amounts of antigen above the transition number/amount of all individuals in the population will result in effective priming for antibody.

I assume this dose dependence of the Th1/Th2 phenotype of the response holds generally. It appears to hold in mice for responses to BCG [23,24] and at least some transplantable tumors [25]. Consider how this generality might allow a universally efficacious form of the low dose vaccination strategy to be developed. Suppose we infect cattle or people with a few BCG below the transition number of all individuals in the population. As the BCG will most likely replicate until contained by immunity, a sustained Th1 response and imprint is expected to develop in all infected individuals. Thus, infection with a very low number of BCG is likely to be universally effective in establishing Th1 imprints [26]. These considerations likely explain the effectiveness of Buddle’s use of a low number of BCG, roughly a million-fold lower than previously employed, in protecting against experimental tuberculosis.

## 8. Our Attempts to Find Parameters That Discriminate Immunity in the Healthy Infected and Tuberculosis Patients

### 8.1. The IgG Isotype Methodology for Monitoring the Th1/Th2 Phenotype of the Immune Response

Our studies in the mouse model of cutaneous leishmaniasis also led us to appreciate the simplicity and incisiveness of a particular methodology. It is known that the Th1/Th2 phenotype of a response is reflected in the isotypes of antibody produced. We found, on examining immune responses in mice to the *L. major* parasite [8,20], to BCG [23,24] and to transplantable tumors [25], that exclusive Th1 responses correlated with undetectable IgG antibodies, predominant Th1 responses, with a small Th2 component, in a low ratio of IgG_1_/IgG_2a_ isotypes among antigen-specific IgG antibodies, and a predominant Th2 response with a high IgG_1_/IgG_2a_ ratio. Employing this “IgG isotype” methodology to indirectly monitor the Th1/Th2 phenotype of a response has three advantages. It reflects systemic immune responses, and it readily allows longitudinal studies in contrast to the frequent procedure of sacrificing mice to assay for the generation of Th1 and Th2 cells. Thirdly, it is simple. These advantages encouraged us to employ it in studies of immunity in human disease.

### 8.2. Human Visceral Leishmaniasis

The pathogen causing this tropical infectious disease is believed to be uniquely contained by a cell-mediated response against the pathogen. The healthy infected and patients were initially identified as being seropositive for parasite-specific antibodies in our field study carried out in a limited geographical area of Ethiopia. Healthy infected individuals express sustained DTH to parasite antigens and may produce some but relatively little IgG antibody, whereas patients expressed little DTH and have higher IgG antibody levels. This disease is rapidly lethal unless treated with anti-parasite drugs. The available drugs show considerable toxicity for patients. The art of clinical treatment has resulted in a short and standard treatment of three weeks. The treatment results in re-expression of DTH and resistance to reinfection, leading us to think treated patients have a Th1 imprint. We examined the IgG_1_/IgG_2_ ratio present among parasite-specific antibodies in healthy infected individuals, in patients at the time of diagnosis and initiation of treatment, and in individuals after treatment. The IgG_1_/IgG_2_ ratio was low in the healthy infected, anticipated from their expression of DTH and the presence of relatively little IgG antibody, expressing a predominant Th1 response. The ratio was high in patients at the time of diagnosis, reflecting presumably a mixed Th1/Th2 or predominant Th2 response. Most interestingly, the ratio in patients following treatment was similar to that of the healthy infected [27]. This presumably explained why treated individuals were resistant to reinfection. Most interestingly, treatment seems to result in modulation of the immune response “backwards” from a humoral towards a cell-mediated mode [27]. This is a type of evolution that does not occur under most natural circumstances, see Figure 1. This treatment of visceral leishmaniasis is ineffective if the parasite is resistant to the drug. It seems likely that decreasing the level of the parasites following drug delivery, and thereby decreasing the antigen load, results in the observed modulation of the immune response. Other evidence in animal systems lead us to believe that antigen dose, and number of CD4 T cells, not only controls the Th1/Th2 phenotype of primary immune responses but of ongoing immune responses. For example, a “stable” mixed Th1/Th2 response in the mouse model of cutaneous leishmaniasis can be modulated to a Th1 mode by partial depletion of CD4 T cells [28].

These studies on how on-going immune responses are regulated are essential for the design of rational strategies of immunotherapy. We shall refer to these studies when considering immunotherapy of tuberculosis.

### 8.3. Immunological Parameters of the Immunity in Tuberculosis Patients and in the Healthy Infected

We quantitated the prevalence of different IgG isotypes among mycobacterium-specific antibodies in sera from patients and healthy infected individuals, who were healthy health care professionals dealing with tuberculosis patients and who tested positive in the skin test. We were inspired by the simplicity and utility of the IgG isotype methodology in our studies in mice of anti-tumor immunity and immunity against *L. major* and BCG. The most striking results came when we examined the IgG_1_/IgG_2_ ratio. This ratio varied over roughly a 1000-fold range among the healthy infected, and over a 100,000-fold range among the patients. The range in the patients overlapped with that of the healthy infected, but the IgG_1_/IgG_2_ value was higher in some patients than the values seen in the healthy infected [3].

This higher IgG_1_/IgG_2_ ratio among some patients was consistent with the idea that these TB patients had a greater Th2 component to their immune response than that of any of the healthy infected; the immune response was similar in kind to that seen in humans ill with visceral leishmaniasis. For convenience, we refer to such patients as having “type 2” tuberculosis.

The most obvious question is why are the patients, with a similar IgG_1_/IgG_2_ ratio as the healthy infected, ill? We have discussed our hypothesis inspired by this enigma elsewhere [3]. Here I state our hypothesis and bullet the main considerations that I think support it.

## 9. The Hypothesis That There Are Two Types of Tuberculosis, Reflecting Two Types of Failure by the Immune System, to Contain the Pathogen

We posit that a sufficiently strong Th1 response contains the pathogen. In type 1 tuberculosis, the predominant Th1 response is too small to contain the pathogen, and so the pathogen load escalates, leading to disease. The Th1 response can contribute to pathology, such as contributing to granuloma formation. In type 2 tuberculosis, the initial Th1 response is undermined as a substantial Th2 component develops, associated with the downregulation of the protective response. Again, the pathogen load is not contained and the Th1 component of the response can contribute to pathology. I suggest this trajectory of the immune response in type 2 tuberculosis is similar to what happens in people who develop AIDS following HIV-1 infection, in patients with visceral leishmaniasis and follows the pattern uncovered by Salvin following the administration of a medium dose of a protein antigen, see Figure 1. The attractiveness of the idea of there being two types of tuberculosis has grown on me over the years. It allows one to relate many observations made on immunity to *M. tuberculosis* to basic observations on immune regulation, and on immunity in some other infectious diseases. It also provides a context for considering how immunotherapy of TB might be achieved. I summarise in bullet form the considerations that I think favour this hypothesis.

The cell-mediated/humoral nature of the long-term anti-pathogen responses among tuberculosis patients has long been recognized as varying over a broad spectrum [4], and our findings support this inference. This is in contrast to the responses in AIDS, cutaneous and visceral leishmaniasis. HIV-1 infected individuals, who control the virus, the elite controllers, either make a Th1, cytotoxic T lymphocyte (CTL) response without production of substantial IgG antibody, or the IgG antibody produced is predominantly of the IgG_2_ class [29], indicating a predominant and stable Th1 response, in contrast to AIDS patients. Cutaneous [15,30] and visceral leishmaniasis patients [31] produce substantial IgG antibodies but express very little if any DTH. The clear demarcation of parameters that distinguish the immunity of most healthy infected and of most patients makes it clear what the purpose of vaccination and immunotherapy must be. The existence of two types of tuberculosis, as proposed and outlined above, may provide an explanation for why patients display such a broad spectrum of cell-mediated/humoral phenotypes [4]. Each “type” of tuberculosis patient, in our interpretation, represents a less broad spectrum. We will consider the treatment of the two types of tuberculosis as separate issues, as they reflect, in our interpretation, different kinds of failure on the part of the immune system.It is well recognized that the dose of non-living antigens administered, or the number of “slowly replicating antigens” employed for infection, dramatically affects the nature and kinetics of the response [32], see Figure 1. This fact is so well known that it is only incidentally documented. However, it is true, for example, of the immune response to mycobacteria, *L. major* parasites, and tumors in mice, as indicated and documented above. This generalization is central to our analysis.A major problem to be faced in thinking of strategies of medical intervention in humans, that is sidestepped in many animal studies employing inbred strains, is the genetic diversity of people. Any attempt to transcend this diversity must, it seems to me, rely on generalizations that hold in most individuals, independently of their genetic constitution. Our studies, described above and showing that it is generally true that the Th1/Th2 phenotype of the ensuing response depends on the number of *L. major* parasites injected into mice, has greatly influenced our thinking. We deliberately chose mice of strains known to be highly disparate in their response to *L. major*. We employ the concept of the transition number in our analysis of responses in people to *M. tuberculosis.*The spread of values in the IgG_1_/IgG_2_ ratio among mycobacterium-specific antibodies in patients over a 100,000 fold range [3] is remarkable. We suppose that the spread in the natural load of mycobacteria causing an infection that leads to illness is rather low and has a much, much less broad range than 100,000. We hypothesize that most of the variation in the IgG_1_/IgG_2_ ratio in patients reflects variation in the value of their transition number for *M. tuberculosis.* As already implied, some individuals with the highest IgG_1_/IgG_2_ ratios above that found in the range of the healthy infected are hypothesized to have a significant Th2 component to their immune response and are designated as having type 2 tuberculosis. These individuals would have the *lowest transition number* in their response to *M. tuberculosis,* below the typical numbers causing infections that lead to disease.We suggest that the anticipated broad range in the value of the transition number for *M. tuberculosis* among individuals can, in the context of the considerations outlined above, lead to a potential explanation for why some infected individuals, though making a “protective’ Th1 response, become ill. Consider those individuals with the *highest transition numbers*, which for convenience we designate as “h”. These individuals will make a rapid and robust Th1 response upon infection with a number of mycobacteria somewhat lower than h, say for convenience h/3. As the tempo of the response is dependent on antigen dose, see Figure 1, the response will only be significantly initiated when the mycobacterial load is, say, h/300. Compare the situation in such an individual with an individual with a transition number of h/100. This individual will have a robust response when the load is h/300, and the response will be initiated when the load is h/30,000.Consider the size of an immune response, S, required to keep the bacterial load in a steady state. In this case, the number of pathogens killed by S and the number generated by replication would be equal. The value of S needed to contain an infection at a particular moment in time thus depends upon the size of the infection at this time. Suppose the two individuals described in 5 above are both infected by the same but low (<h/30,000) load of *M. tuberculosis*. The individual with a transition number of h/100 will initiate making an immune response when the bacterial load reaches h/30,0000, and the size of the response needed to contain the infection will only have to contend with the load arising from h/30,000 mycobacteria. In contrast, the individual with a transition number of h will start making a robust response when the bacterial load reaches h/300, so the response this individual needs to contain the infection will therefore be much larger. It seems plausible to me from these considerations that the individual with a transition number of h/100 will contain the infection, but the individual with a transition number of h will often not, and so suffer disease. I think it helpful at this stage to summarize the overall hypothesis.

We anticipate the transition number for *M. tuberculosis* varies greatly among the human population. Those with the lowest values, with a value lower than the infective dose, will, upon infection, generate a rapid Th1 response that, with time, develops a substantial Th2 component, associated with a downregulation of the protective Th1 response, and so containment of the pathogen is undermined. These patients have type 2 tuberculosis. Their IgG_1_/IgG_2_ ratio when diagnosed with disease is large. Most infected people have intermediate transition numbers, somewhere a bit above the infective dose associated with a primary infection. They make a moderately rapid and sustained, predominant Th1 response, and this response contains the pathogen before it has reached high levels. They are the healthy infected and their IgG_1_/IgG_2_ ratios are between small and intermediate. Other individuals have very high transition numbers. These individuals only start making a response later when the bacterial load is high. Containment would require a much larger Th1 response, and this cannot usually be reached. These individuals have type 1 tuberculosis. Their IgG_1_/IgG_2_ ratios are low to intermediate.

I would like to describe two facts bearing on my thinking about the plausibility of type 1 tuberculosis. I noticed, during our studies on the mouse model of cutaneous leismaniasis, that “resistant” mice, injected in the foot with a million parasites, make a sustained Th1 response that in time results in parasite containment. The resolution of the infection takes considerable time, a number of weeks. A standard practice in such studies is to measure the increase in foot thickness, often more than a doubling, as a measure of the gravity of the lesion. It takes a week or two for the lesion to substantially develop, and some further weeks for it to resolve. I think there are likely different processes contributing to lesion size, but surely among these is DTH to parasite antigens present in the foot, and the inflammatory reactions consequent to tissue damage. I could not help but reflect that these mice were said in the literature to be resistant; however, if the lesion was in the lung, rather than the foot, the mice might have suffered more obvious pathology. I think there may be a related observation concerning tuberculosis patients from before the antibiotic era. Some patients in sanatoria were found to “spontaneously cure” [33], a process that may well be similar to the eventual disappearance of foot lesions in “resistant” mice, that generate a sustained Th1 response.

I had read, early on in my interest in tuberculosis, that tuberculosis is a “Th2” disease, and that Th1 responses are protective. The existence of spontaneous cures made me skeptical that all cases of tuberculosis could be regarded as a Th2 disease. Responses rarely, if ever, evolve, under natural conditions, from a mixed towards an exclusive Th1 mode, see Figure 1. It seems natural that the spontaneous cures observed long ago were due to a protective Th1 response catching up with and overwhelming the mycobacteria in patients with type 1 tuberculosis. I argue below that it may be possible to encourage the equivalent of such spontaneous cures by immunotherapy.

## 10. Strategies of Immunotherapy

I have suggested here and elsewhere [3] that there are two kinds of tuberculosis, reflecting two distinct types of failure by the immune system. I propose that any immunotherapy should initially involve an assessment of whether a patient has type 1 or type 2 tuberculosis, primarily based on their IgG_1_/IgG_2_ ratio among mycobacterium-specific antibodies, as the envisaged treatments would be different.

### 10.1. Type 2 Tuberculosis

I have suggested above that the nature of the failure to contain the pathogen is similar in visceral leishmaniasis and type 2 tuberculosis. Visceral leishmaniasis is treated by giving a short course of drugs that kills the parasite, resulting in a modulation of the Th1/Th2 response to a protective, Th1 mode, a treatment that correlates with the IgG_1_/IgG_2_ ratio becoming small. I suggest a short antibiotic treatment, resulting in a low IgG_1_/IgG_2_ ratio, would similarly constitute a cure of type 2 tuberculosis. We have recently suggested a similar treatment for HIV-1 infection [34]. Such a treatment of tuberculosis patients would not only be effective but likely much shorter than current treatments, and in this sense be more practical. As we have discussed in the case of treatment of HIV-1 infections, further treatment with drugs, once a protective response has been optimally harnessed, is expected to further lower the antigen load and in time undermine the protective response in most people. Current prolonged antibiotic treatment of type 2 patients likely undermines the protective response, just as prolonged anti-retroviral therapy is believed to undermine the protective response against HIV-1 [33].

### 10.2. Type 1 Tuberculosis

I proposed above that people with type 1 tuberculosis have, relatively speaking, very high transition numbers, and their immune system only responds to relatively high amounts of mycobacterial antigen. The size of the Th1, protective response is likely limited by the amount of antigen available. Increasing the antigen available, by administering antigen, without increasing the pathogen load, is envisaged to be a means to increase the protective Th1 response and turn the tide to the benefit of the patient. It would be essential to longitudinally monitor the IgG_1_/IgG_2_ ratio to ensure that the response is not being pushed to acquire a significant, detrimental Th2 component. Current antibiotic treatment reduces the antigen load and so is expected to undermine the little protective immunity that is present but insufficient in type 1 patients. This analysis perhaps explains why current antibiotic treatment must be so long to be efficacious, as it must eliminate every last pathogen. It is possible that the proposed treatment of type 1 tuberculosis can be relatively short, as the protective immune response is being modulated to become stronger, resulting in a decreased level of pathogen, and so a Th1 response of decreased size could contain the pathogen. In this case, it would be more practical and provide less opportunity for the pathogen to develop antibiotic resistance.

## 11. Revisiting Koch’s Treatment and Facing the Problem Posed by Antibiotic Resistance

It is fascinating to read some of the original papers on Koch’s treatment [35,36,37,38]. The two outcomes that are documented appear to follow from the framework we have developed here. Thus, we anticipate that his treatment of administering mycobacterial antigens would, in patients with type 1 tuberculosis, increase protective immunity and so facilitate a “spontaneous cure.” We anticipate that similar treatment of patients with type 2 tuberculosis would foster an increase in the detrimental Th2 component of the immune response and exacerbate disease and possibly hasten death; treatment may drive the immune response towards a Th2 mode, leading to miliary tuberculosis.

We now consider the potential effects of these proposed treatments for tuberculosis in the context of the prevalence of drug resistant pathogens. We consider first the potential of effective immunotherapy on the generation of drug-resistant pathogens. Secondly, we consider how the ideas developed here might aid in the treatment of patients whose pathogens are resistant to the available drugs.

It is well acknowledged that drug resistance has become prevalent in large measure because directly observed therapy must be carried out for months to be effective, a course that sometimes fails to be completed, giving rise to reservoirs of pathogens resistant to one or more of the antibiotics used for treatment. I understand that standard treatment, to be effective, must kill virtually every last bacterial pathogen in the patient to be effective. This is likely why standard and effective treatment must be so long. I suggest that much shorter treatment may be possible if the treatment is designed to harness and effectively harnesses the patient’s own protective immunity. This is likely true of the proposed treatments of both type 1 and type 2 tuberculosis.

An expectation of our framework is that patients with type 1 tuberculosis, administered antigen in conjunction with standard antibiotic treatment, will have considerably greater and sustained production of IFN-gamma by their peripheral lymphocytes in response to mycobacterial antigens than patients administered antibiotics alone. This expectation is readily testable. Moreover, we anticipate, if our proposal is valid, that some patients will “spontaneously cure” in a similar way that occurred in the pre-antibiotic era, as their level of Th1 cells is sufficient to kill *M. tuberculosis* faster than they are generated through bacterial multiplication. If such a treatment were found to be efficacious, it may be possible to treat patients with antigen alone that are infected with a problematical multi-drug resistant strain, so recapitulating Koch’s vision, but in a much more broadly considered context.

The envisaged immunotherapeutic strategy of type 2 tuberculosis does not require the elimination of the pathogen, but just to decrease its multiplication sufficiently to reduce the pathogen and the antigen load such that the response is modulated into a protective mode. It seems to me this is a less stringent requirement.

## 12. A Case for Optimism

I must say that I am surprised to read in review after review of the many ways the tuberculosis pathogen has evolved to avoid innate defense and adaptive immunity. I have always told my graduate students not to be deterred by this. After all, about 90% of people infected by the pathogen resist it [19], and I suggest this is what they should keep in the forefront of their minds.

## 13. Concluding Comment

I have shared for many years the feeling that the findings on immunity in individuals infected by *M. tuberculosis* are difficult to fit into a broad framework. It has taken me years to integrate both qualitative and quantitative ideas from other disciplines into a framework where many observations on TB immunity fall into place. I hope readers can share my excitement. I hope the ideas presented here might help in controlling the ravages of this disease.

## Figures and Tables

**Figure 1 antibiotics-11-00371-f001:**
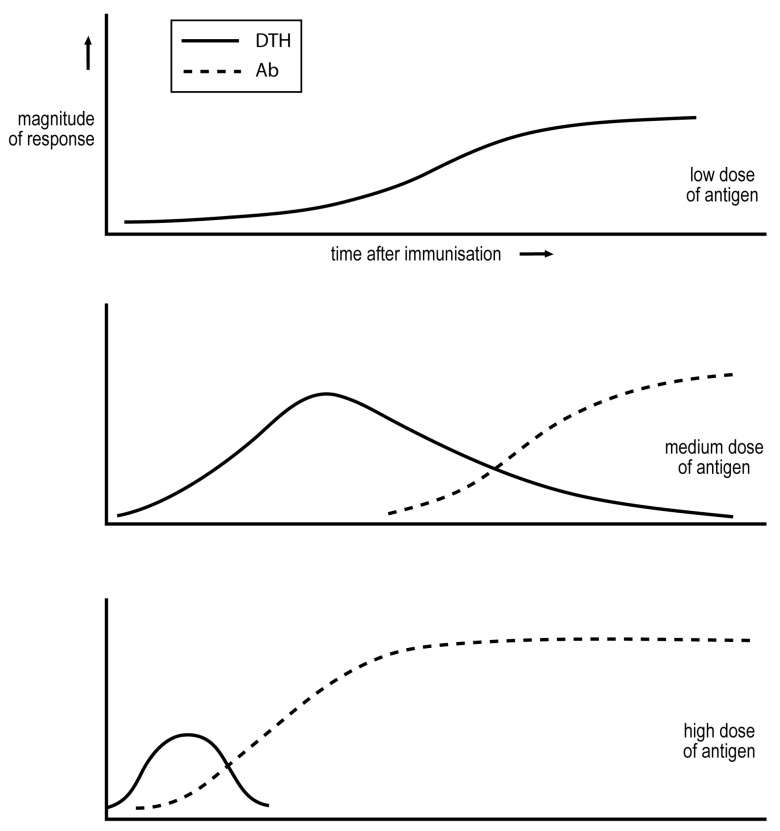
Dependence of cell-mediated/IgG antibody nature of response on dose of antigen and time after antigen impact.

## Data Availability

Not applicable.

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
