# Peer review of "Facing the Increased Prevalence of Antibiotic-Resistant *M. tuberculosis*: Exploring the Feasibility of Realising Koch’s Aspiration of Immunotherapy of Tuberculosis"

_antibiotics, 2022, doi:10.3390/antibiotics11030371_

Round 1
Reviewer 1 Report
After careful review of this paper, although the theoretical foundations of the immune response from Mycobacterium tuberculosis are complete, this paper is written more as an academic seminar, with the author's personal style and "apology" to readers.
This manuscript reconsider Koch’s vision in the treatment of tuberculosis to better understand the overall protective immunity, in order to improve it in terms of efficacy and efficiency. It is an original topic considering the IgG1/IgG2 ration among mycobacterium-specific antibodies, as the envisaged treatment in order to overpass antibiotic resistance. Administration of antigens before antibiotic treatment in patients with tuberculosis infection can boost a stronger Th1 immune response, shorter in period so that the pathogen could not have the time to develop antibiotic resistance.
The author mentioned that he has little knowledge about the field of drugs used to fight tuberculosis (lines 596-597) but he proposes that it might be possible not to administer antibiotics in the treatment of patients with type 1 tuberculosis. The immunotherapeutic strategy proposed maybe helpful but an overall consideration of the tuberculosis infection based on the therapeutic schemes used by clinicians, has to be made.
Major revision is needed so that these interesting opinions can be addressed to the scientific community as a research paper.
Author Response
In considering all three reviews of my manuscript I decided to completely rewrite it to focus on matter more closely related to the title. Nevertheless, I have still tried to cover ground from basic immunology, infectious diseases and immunity against M tuberculosis that I regard as central to the ideas on immunotherapy that I am suggesting. With regard to this reviewer in particular, I addressed how immunotherapy of TB can be combined with current DOT. I do hope the the revised manuscript is found to be of interest and suitable.
Reviewer 2 Report
Dr. Bretscher wrote an interesting thought article on understanding why immunotherapy fails for M. tuberculosis and how to overcome those challenges that include the failure of Koch's efforts.
Although the review in itself encourages a conversation of the immunotherapy challenges, the article fails to present relevant information about M. tuberculosis immunotherapy failures. The author has taken inspiration and summarized the current observations made in immunotherapy experiments taking place in the field of cancer, leishmaniasis, etc. that already is trying to address the failures raised due to genetic diversity of patients, the Th1 vs. Th2 responses effect on the pathogenesis, and the co-evolution of pathogens. These are very generalized observations that do not provide a new idea to solve the current problem but do answer the observation that Koch's experiments saw.
Further, the author has ignored the importance of choosing the right antigen, and the release kinetics of the antigen that affects the efficacy of the immunotherapies.
Author Response
In considering all three reviews of my manuscript I decided to completely rewrite it to focus on matter more closely related to the title. Nevertheless, I have still tried to cover ground from basic immunology, infectious diseases and immunity against M tuberculosis that I regard as central to the ideas on immunotherapy that I am suggesting. With regard to this reviewer in particular, I would like to explain a difficulty in talking across silos. It is my understanding that the reviewer feels a weakness is that I do not address what particular antigen/antigens are important in the immunity raised in a healthy contact or a patient in determining whether the immunity is protective or not. He/she feels the diverse comments on immunity in related fields, such as cancer and the leishmaniases, are somewhat removed from the issues at hand, particularly in this context. I wish we could have a conversation on this. One feature of immunity to complex antigens, such as diverse pathogens, is that they are often "coherent", i.e. the Th subset to which the CD4 T cells belong, specific for diverse antigens, is the same. This "coherence" is not a generally recognized feature of immune responses. In my mind, I think in terms of getting the correct type of immunity to all the antigens, and then we will have the correct type of immunity to the target antigens. In this case, it is not necessary to define the target antigens. I have added a section in the revised ms addressing this concern. I hope the revised manuscript at least partly covers these concerns of the reviewer. I do hope the the revised manuscript is found to be of interest and suitable.
Reviewer 3 Report
In the manuscript entitled “Exploring the feasibility of realizing Koch’s aspiration of immunotherapy of tuberculosis in the face of the increasing prevalence of antibiotic-resistant M tuberculosis” the author has explored the suitability of rationalizing immunotherapy to overcome the problems posed by antibiotic resistance of the pathogens. Below are the suggestions to improve the manuscript.
- Lines 32-33: In this case, the critical questions are, what kind of immunity can contain the 32 pathogen and how can such immunity be attained?
- On the page 2 of the manuscript, while discussing about the Successful vaccination, the author only focuses on the humoral immune response (antibody responses). What about the cell-mediated immune responses? A successful vaccination elicits a robust humoral and cell-mediated immune responses. The author should cover the contribution of the cell-mediated immune responses towards the control of the disease.
- Lines 58-59: It is less clear what the immunological correlates of protection are, upon infection with M tuberculosis
- Line 183: ………in vivo
- Line 470: ………. required to generate sufficient signal 1 for activation is similar or less than the level required…….
- The author discusses and comes up with hypotheses to treat tuberculosis and later to the scenario of antibiotic resistance. Did the author test these hypotheses experimentally? Where is the data to either support or refute the hypotheses?
Author Response
In considering all three reviews of my manuscript I decided to completely rewrite it to focus on matter more closely related to the title. Nevertheless, I have still tried to cover ground from basic immunology, infectious diseases and immunity against M tuberculosis that I regard as central to the ideas on immunotherapy that I am suggesting. With regard to this reviewer in particular, I did address that cell-mediated immunity, not antibody, is protective, and how my strategy is informed by this view. Of course I did justify in my previous ms, and in the revised version, the proposed immunotherapeutic treatment. The call for tests of the proposed therapy, before it should be considered, is not very helpful for discussion that leads to new ideas. I do hope the revised manuscript is found to be of interest and suitable.
Round 2
Reviewer 1 Report
The author proceeded to an extensive formulation of the text with several bibliographical references on the biochemical mechanisms of immunization in tuberculosis.
It is a text in which there are several references to the personal work of the author that could be omitted.
For the rest of the text, i think it's okay.
Author Response
Thank you. I feel the references to the work of my lab really is necessary to validate the statements made.
Reviewer 2 Report
The author in this paper proposes an immunotherapy strategy that could hypothetically address and overcome the challenges that Koch's immunotherapy faced. The author has improved the connection between the observation he observed in the leishmaniasis study and how it translates into the TB model.
I would like to have a clarification from the author on is the difference between "the immunity of patients" versus "immunity in the healthy infected". Are these two different groups of patients infected by different pathogens?
The author has written in their reply that "it is not necessary to define the target antigens", I slightly disagree with this comment. The author of this manuscript has focused on studies where mice were vaccinated with attenuated bacteria. As per the question raised in this manuscript, lowering the bacterial dose in such whole bacteria vaccines could potentially modulate the patient's immune system toward sustained Th1 immune response.
But as we know that BCG vaccine is not given to individuals in multiple countries due to its variability, therefore alternate vaccines that utilize an antigen from the pathogen have become much more relevant to protect individuals in such countries where BCG is not administered. Therefore, identifying proper target antigens and looking into the release kinetics of such antigens in the patient's body is extremely crucial for developing resistance against MTb, in patients vaccinated with purified antigens rather than whole bacteria.
With respect to this paper, the arguments presented by the author especially on identifying two types of tuberculosis pathogenesis provide a good hypothesis on why we observe variability in patients when we use whole bacteria vaccines like BCG including Koch's observation in his investigative study in TB patients.
Slight editing in the grammar and sentence structures is required as they don't slow smoothly, therefore, losing the impact.
Author Response
Thank you. I have responded in the text to the question : what is meant by the difference between "the immunity of patients" versus "immunity in the healthy infected". Are these two different groups of patients infected by different pathogens? The sort answer is that the infected were detected as being seropositive for L donovani, and this was a field study carried out in Ethiopia in a limited geographical area.
I have commented in the text that, though it it is not necessary to define target antigens, this does not mean vaccination with purified antigens should not be considered if found to be effective.
Reviewer 3 Report
The author has addressed the concerns adequately.
Author Response
Thank you.